# Modulating Glycoside Hydrolase Activity between Hydrolysis and Transfer Reactions Using an Evolutionary Approach

**DOI:** 10.3390/molecules26216586

**Published:** 2021-10-30

**Authors:** Rodrigo A. Arreola-Barroso, Alexey Llopiz, Leticia Olvera, Gloria Saab-Rincón

**Affiliations:** Departamento de Ingeniería Celular y Biocatálisis, Instituto de Biotecnología, Universidad Nacional Autónoma de México, Cuernavaca 62271, Mexico; rodrigo.arreola@ibt.unam.mx (R.A.A.-B.); alexey.llopiz@ibt.unam.mx (A.L.); leticia.olvera@ibt.unam.mx (L.O.)

**Keywords:** transglycosidation, hydrolysis, contact-residues, amylase, glucanotransferase, coevolution, enrichment-factor, specificity

## Abstract

The proteins within the CAZy glycoside hydrolase family GH13 catalyze the hydrolysis of polysaccharides such as glycogen and starch. Many of these enzymes also perform transglycosylation in various degrees, ranging from secondary to predominant reactions. Identifying structural determinants associated with GH13 family reaction specificity is key to modifying and designing enzymes with increased specificity towards individual reactions for further applications in industrial, chemical, or biomedical fields. This work proposes a computational approach for decoding the determinant structural composition defining the reaction specificity. This method is based on the conservation of coevolving residues in spatial contacts associated with reaction specificity. To evaluate the algorithm, mutants of α-amylase (*TmAmyA*) and glucanotransferase (*TmGTase*) from *Thermotoga maritima* were constructed to modify the reaction specificity. The K98P/D99A/H222Q variant from *TmAmyA* doubled the transglycosydation/hydrolysis (T/H) ratio while the M279N variant from *TmGTase* increased the hydrolysis/transglycosidation ratio five-fold. Molecular dynamic simulations of the variants indicated changes in flexibility that can account for the modified T/H ratio. An essential contribution of the presented computational approach is its capacity to identify residues outside of the active center that affect the reaction specificity.

## 1. Introduction

Enzymes are accelerators of chemical reactions that occur in living cells, which also work in vitro, making their use in the laboratory, in medical applications, and in industry possible [1,2,3]. Tailoring an enzyme’s ability to carry out specific reactions is one of the greatest challenges that must be met in order to move on to a more sustainable biocatalysis process [4]. In this sense, directed evolution has proven to be a valuable strategy for evolving functions, with the limitation of requiring extensive screening efforts, in order to find an improved biocatalyst [5,6]. De novo design has shown impressive improvements over the last two decades in the development of energy functions for directing the design of proteins [7,8,9]. However, the subtle changes that confer the necessary dynamics for catalysis have not yet been determined [10,11]. Last year, an enormous breakthrough was made in the implementation of artificial intelligence tools that predict the 3D structure of proteins. Alpha-Fold surpassed the performance obtained so far by any other structure prediction method in the CASP protein modeling competition using this approach. This year, the modeling of protein structures from humans and 20 other genomes through the use of artificial intelligence is in progress using one of the most powerful supercomputers [12,13].

The above-mentioned strategies have highlighted the importance of exploiting the structural and functional information accumulated through thousands of years of evolution. It is acknowledged that residues outside the catalytic site and their contacts can play essential roles in protein structure and function [14,15]. Exploration of these residues is required to continue improving the functional facet of protein design. Nature has evolved new enzymes for novel reactions from pre-existing ones, giving rise to shared structures and sequence motifs in proteins with distinct functions [16,17,18,19]. Functional insights have already been attained by comparing features between evolutionarily related but functionally different enzymes. Although such analysis has focused on the catalytic site, we are starting to understand that the investigation should extend outside the protein core. 

Directed evolution and massive gene sequencing of mutational libraries (e.g., deep sequencing) have highlighted the relevance of residues outside the catalytic site for protein function. In the case of deep sequencing, it has also revealed pairs of residues that are in contact, making the reconstruction of 3D protein structures possible by analyzing the effects of concurrent mutations on their activity [20,21]. Structure reconstruction based on deep sequencing data parallels the analysis of residue contacts, in which specific patterns characterize different protein families [22,23]. These patterns are frequent inputs for algorithms predicting 3D protein structures [13,24], identification of members belonging to functional families [25,26], the study of coevolution through protein domains [27], and reconstruction of novel sequences whose structure and function matches an already existing family [28]. 

A more direct attempt to exploit residue contacts in order to modify function is the analysis of residue covariation in a multiple sequence alignment. The amino acids identified as covariates are targets for mutagenesis, and thus, investigations have identified residues whose substitution impairs function [29,30]. The advent of Alpha-Fold makes structural models available like never before, with similar reliability to those obtained from X-ray crystal diffraction of many proteins for which we only have the amino acid sequence [13]. This fact encourages us to explore currently accessible X-ray structures to compare and identify the information we may be able to draw from Alpha-Fold.

The CAZy database (Carbohydrate Active Enzymes database [31,32]) glycoside hydrolase family 13 of proteins (GH13) shares a (β/α)_8_ TIM barrel core as a catalytic domain and at least two different domains termed B and C [33]. This enzyme family includes enzymes acting on α-(1→4) bonds in glucose polymers and oligomers that differ in their reaction specificity. In both subgroups, the enzymes break the *O*-glycosidic bond of polysaccharide chains in two parts. An intermediate is formed, with one sugar fragment covalently bound to the protein through the anomeric carbon of its reducing end [34]. This double-displacement mechanism ends with the transfer of the sugar moiety to an acceptor molecule, which defines the type of reaction that takes place. Those members with hydrolytic activity transfer the glycoside moiety to water, while others that are predominantly transferases, transfer it to another sugar [35,36,37,38]. 

The balance between transglycosylation and hydrolysis must rely on many factors inherent to the protein, such as its sequence, 3D structure, acid-baseproperties of its critical residues, hydrophobicity at its active site [39,40], flexibility, and dynamics. The change of the acid-basecatalytic residue pKa during the reaction is crucial for both reactions. However, the pKa increase of the acid-baseresidue may be more relevant for the hydrolysis than for the transglycosylation reaction, since proton removal from a water molecule is energetically more demanding. Recently, Geronimo et al. [41] reported a molecular dynamic simulation at a constant pH for beta-glycosidase (bglc) from *Hypocrea jecorina*. Using this approach, the authors observed that the usual rearrangement of the active center (residues R169, Y204, and W237) is different if acceptor nucleophiles like cellobiose or glucose are present. They found that as a result, the pKa value for the catalytic residue (E441) is lower in the presence of these sugars than in that of water. Their results suggest that protein specificity towards hydrolysis or transglycosylation may be associated with protein dynamics and flexibility, which in turn, can be influenced by the presence or absence of acceptor nucleophiles. Additionally, many loops located in the vicinity of the active site play an essential role in stabilizing the transition state for many amylases [42].

Multiple reports describe the modification of amino acid residues for increasing the transglycosylation/hydrolysis (T/H) ratio of amylolytic enzymes [35,43,44,45]. The factors considered in the selection of amino acids for this purpose include the location of flexible regions associated with internal water transport [46,47], shifts in acid-baseresidue dynamics [47], and sidechain conformational changes of residues near the active center [48]. Some other investigations have intended to modify the specificity of the acceptor site toward diverse organic molecules [49] and have succeeded in changing the T/H ratio; nevertheless, the strategies explored have generally focused on punctual mutations near the active site identified by multiple sequence alignments (MSA) of a few glycosidases—presumably with high specificity towards hydrolytic or transfer reactions [43,45,50,51,52]. 

The relationship between coevolving residues and protein specificity has been evaluated in the aquaporin family in order to find the residues associated with water or glycerol transport [53]. Additionally, coevolving residues associated with protein specificity were evaluated in twelve families of proteins [54]. However, in many cases, the coevolution of residues is hard to detect because of (1) the presence of changing compensatory network mutations, (2) the significant dependence of covariations on evolutionary distances, (3) the number of proteins in an MSA, and (4) the quality of alignment in the coevolving residues’ environment. 

Here, we propose an algorithm for the differential analysis of contact patterns among evolutionarily related α-glycosidases with two distinct reaction specificities. Contact maps, 2D representations of 3D structures, were used to compare the enrichment of each pair of residue contacts. The proposed approach has the advantage of reducing the dimensionality of the system [55,56] in order to identify the elements driving the specificity between hydrolysis and transglycosylation in the glycoside hydrolase family 13. Understanding the molecular determinants of protein specificity could contribute to the development of enzymes for glycosynthesis (e.g., adding a sugar moiety to an organic molecule) and design of enzymes with desired increases in hydrolytic or transglycosidic specificity. We used two model enzymes to validate our predictions: α-amylase (*TmAmyA*) and glucanotransferase (*TmGTase*), from a hyperthermophilic bacterium *Thermotoga maritima*.

## 2. Results

We analyzed residue-residue contacts in 14 structures (Dataset 1, four transglycosidases and 10 hydrolases belonging to the GH13 family bound to acarbose, Appendix A). The selection of structures was based on the availability of structures bound to acarbose, a transition state analog. It is important to mention that all proteins were monomeric to the best of our knowledge. Inclusive *TmGTase*, whose crystal structure suggests a dimeric protein, has been reported as being in an equilibrium of 90% monomers, 10% oligomers when it is in solution [57]. Thus, the possibility that the determinants of specificity detected in this way were due to the oligomeric interphase contribution was ruled out. After comparing the residue contacts of enzymes with transglycosidic activity against these hydrolytic reactions, we identified preferences between the groups for different amino acids when forming pairs in each residue-residue contact. These results agreed with the notion that enzymes work under selective pressure, and that residues coevolve to create the residue-residue contacts that maintain structure and function. We identified contacts in which some amino acids were frequently present in either hydrolases or transferases, and underrepresented in the other group. We measured and expressed these preferences using the enrichment factor ($\Delta f_{aa}^{ij}$) described in the Methods section (Equations (1) and (2)).

### 2.1. Homology Model of TmAmyA

The homology model of *TmAmyA* was constructed using the crystallographic structure of the amylase from *Thermotoga petrophila* (PDB ID 5M99, resolution 1.96 Å) as a template. This model excludes thirty extra residues at the N-terminus of *TmAmyA* not present in the crystallized amylase from *Thermotoga petrophila*, and which do not belong to the core domains of GH13 enzymes. The 504 remaining residues have 98.4 % sequence identity, showing only six substitutions. The “Structure assessment” tool [58] from the Swiss-Model server was used to validate this model. The QMEAN value was 1.06; a similar value was obtained for the structure used as the template. The MolProbity Score had a value of 2.96, and Ramachandran Favoured was 96.63%. Additionally, the validation with ProSA-web (*Z*-Score of −9.95) had a value similar to other proteins of a similar size. VERIFY3D (99.21% of the residues had an average 3D-1D score ≥0.2) suggests that the model is of adequate quality for the study presented in this work. 

### 2.2. Protein Classification as Hydrolases or Transglycosidases

The residue contacts in *TmGTase* (PDB ID: 1LWJ) were compared with those of the structures within Dataset 1 and Dataset 2 (two tranglycosidases and 12 hydrolases of the GH13 family not containing acarbose, different from Dataset 1; Appendix A). The amino acids in all residue contacts shared with 1LWJ were qualified as hydrolytic if $\Delta f_{aa}^{ij}<0$ (the amino acid was more frequent in hydrolases) and transglycosidic if $\Delta f_{aa}^{ij}>0$ (the frequency of the amino acid was higher in the transferases). After qualifying all the amino acids in the residue contacts, contacts formed with a pair of residues deemed transglycosidic were designated transglycosidic; those formed by two hydrolytic residues, as hydrolytic. As expected, the relation of hydrolytic/transglycosidic contacts in each protein classified all the enzymes correctly as hydrolases or transferases in Dataset 1 (Figure 1a), which was used to calculate the $\Delta f_{aa}^{ij}$ values by the algorithm. The $\Delta f_{aa}^{ij}\;$obtained from Dataset 1 was sufficient to classify all the enzymes in Dataset 2 according to their primary function (Figure 1b), despite its incomplete sampling of the sequence space, as suggested by the residues that remained unclassified using the obtained $\Delta f_{aa}^{ij}$ (data not shown). The lack of complete information about the reaction specificity of some of the enzymes, and the fact that not all residues identified are involved in determining the reaction specificity, could account for this discrepancy. Nonetheless, the classification of enzymes based on residue-residue contacts seems to indicate the presence of subgroups whose transglycosidic/hydrolytic activity would be appealing to characterize in future studies to see if the classification correlates with their grouping in this graphic (Figure 1a,b). During the process of this work, new members of the GH13 family were added to the CAZy database. Addition of the new structures resulted in Dataset 4, which contained a more significant number of enzymes (31 transglycosidases and 40 hydrolases belonging to the GH13 family, comprising all the characterized enzymes active on 1,4-α-bonds according to the CAZy database, which includes all enzymes in Dataset 1 and Dataset 2), and where it was also possible to discriminate enzyme function using enrichment factors (Figure 1c). Still, the structure, corresponding to an α-amylase from bacilli (PDB ID:1QHO) was separated from the rest of the α-amylases, and was closest to the transglycosidases, while the transglycosidases closest to the hydrolases were not from bacilli. As many transglycosidases are of bacilli origin in Dataset 4 (Appendix A), this suggests that the enrichment factors in Dataset 4 reflect the specificity and phylogenetic origin of the enzymes.

Measuring contact conservation derived a helpful parameter for phylogenetic analysis. The contact conservation score was calculated as the fractional number of times that each particular contact in a reference protein was present in all the enzymes in the dataset. We computed the contact conservation scores using different enzymes as the reference each time. The residue contact conservation score of each enzyme was plotted against its equivalent contact with other enzymes. The plot for each pair of enzymes produced a correlation coefficient that indicates the degree of evolutionary relatedness between the enzyme pair. Figure 2 shows the correlation between all the proteins in the Dataset 3 (39 enzymes belonging to the GH13 family, one to the GH97 family and one belonging to the GH31) and *TmAmyA*. The lowest correlations correspond to the enzymes identified with the PDB ID: 2ZQ0 (family (GH97) and 3W37 (family GH31)—an expected result since these enzymes did not belong to the GH13 family. Thus, this correlation value could be used as a contact similarity coefficient to discriminate between members of different glycoside hydrolase families. 

The contact similarity coefficient correlated with the structure DALI Z-value (Figure 3a), which showed a relationship with sequence identity [59]. However, its resolution for identifying differences between proteins was greater than that of pairwise sequence distance, for which the values plateaued for the proteins under consideration (Figure 3b). The contact similarity coefficient showed more sensitivity to enzyme similarity than the sequence similarity parameters. Despite the structural and sequence difference between *TmAmyA* and the rest of the enzymes, the contact similarity coefficient was able to identify *Thermotoga maritima* GTase (*TmGTase*) as its closest structure. The contact similarity coefficient also grouped some GH13 subfamily members (subfamily 2: 4JCL, 2CXG, 1UKQ; subfamily 13: 2YOC, 2FHF; subfamily 8: 3AMK, 3AML) and identified outsiders (GH97: 2ZQ0, GH31: 3W37; Appendix A). Although it misplaced structures, such as 3K8M, that should have been close to *TmAmyA*, this could be due to additional domains not present in the GH13 family [60]. As the comparison of the dataset with a single structure yielded such identifications, the comparisons between the whole dataset may provide a tool for performing automated classification of enzymes when performing the analysis of domains.

### 2.3. Modification of Reaction Specificity in the α-Amylase A from Termotoga maritma

One of the goals for increasing transglycosidic activity in glycoside hydrolases is to create glycosynthetic biocatalysts. We decided to use the *Thermotoga maritima* α-amylase AmyA (*TmAmyA*) as a model of a hydrolase. This enzyme offers as advantages a high thermal stability, with an optimal temperature above 80 °C, and an efficient saccharifying starch hydrolysis pattern, associated with a considerable transglycosylation activity [43]. Mutations for increasing the tranglycosidic activity of *TmAmyA* should substitute residues more frequently found in hydrolases ($\Delta f_{aa}^{ij}<0$) by others more frequently present in transferases ($\Delta f_{aa}^{ij}>0$). The criterion for selecting residues for mutagenesis was if the original residue in *TmAmyA* had an enrichment value ($\Delta f_{aa}^{ij}$ ) below −0.2 (residues enriched in hydrolases) or if the $\Delta f_{aa}^{ij}$ for potential candidate residues was above 0.2 (enriched in transferases), while the *TmAmyA* residue had a $\Delta f_{aa}^{ij}\leq 0$. A valuable result from this approach was the identification of targets for mutagenesis beyond the catalytic site, whose relevance in terms of their specificity and activity has been shown by directed evolution [41]. The exploration of such sites is limited by the availability of selection or high-throughput screening methods; thus, restricting the sequence space that is to be evaluated becomes paramount. 

When selecting a residue to mutate, preference was given to residues not in the catalytic site that were conserved among the enzymes used (in more than 10 out of 14 enzymes). As a proof of principle, we selected a contact complying with these restrictions to produce the variant K98P/D99A double mutant (K98 with $\Delta f_{aa}^{ij}$ = 0 against P98 with $\Delta f_{aa}^{ij}$ = 0.45 on one side, and D99 with $\Delta f_{aa}^{ij}$ = 0.00 against A99 with $\Delta f_{aa}^{ij}$ = 0.4 on the other side; Figure 4a). These residues were located in a very long loop joining the β2-strand and helix-2 in the TIM barrel, which is not part of any of the highly conserved regions in the GH13 family. K98P/D99A substitutions represent a drastic change of physicochemical properties (Appendix A). However, a Pro residue would favor the turn observed in the structural model at the targeted position, and substituting both residues would change the $\Delta f_{aa}^{ij}$ to a positive value, which is desirable in transglycosidases (Figure 4a; for more $\Delta f_{aa}^{ij}$ values at this position see Appendix A). These mutations were evaluated in the wild-type background and combined with the H222Q mutation near to the catalytic site previously reported to increase the transglycosylation/hydrolysis ratio [28].

In addition to its impact in the synthesis of alkyl-glycosides, the alcoholysis reaction (i.e., reaction of a sugar with an alcohol to produce an alkyl-glycoside) can be used as a proxy for the transfer reaction to other sugars [30]. This approximation was used because *TmAmyA* transglycosydation products do not significantly accumulate during the predominant hydrolysis reaction. We thus performed the depolymerization of starch in the presence of 10% 1-butanol to obtain butyl-glycosides as products and evaluated the alcoholysis yield in the different *TmAmyA* variants, as previously described [43]. As a result, the double mutant K98P/D99A succeeded in increasing the T/H ratio by a factor of 1.17 (Figure 5a). The mutations produced by the enrichment factors had a more significant impact on hydrolytic activity, with an approximately 25% reduction (Table 1). On the other hand, the alcoholysis yield was 17% lower over the wild-type background, while having a near 10% increase compared to the H222Q variant. 

### 2.4. Increasing Hydrolase Activity in the 1,4-α-Glucanotransferase from Thermotoga maritima

As a proof of concept, we wanted to know if the enrichment factor could be used in the other direction—to turn an enzyme which is mainly a transferase into a hydrolase. For this purpose, we selected the GTase of *T. maritima* (*TmGTase*). In this case, we analyzed residue pairs instead of residues within a contact pair, as the analysis of residue pairs was an efficient parameter for classifying GH13 enzymes according to their function (Figure 1). The contact pairs were better than the individual residues for classifying enzymes according to their reaction specificity. For this reason, the use of contact pair enrichment, instead of the enrichment of individual amino acids within each pair, should increase the chance of selecting substitutions that transform a transglycosidase into a hydrolase. This would also ensure the selection of pairs more representative of those found in the transglycosidic and hydrolytic enzymes. Additionally, we included the parameter of betweenness centrality—a measurement of the role of a node in transferring information within a network [61]—to restrict our search of mutation sites further. This centrality parameter is calculated as the sum of the fraction of the paths between all pair nodes *i* and *j* containing the node *v*, distinct from *i* and *j* [62]. This parameter is reported as a measure of the importance of specific amino acid residues for the structure and function of proteins [63]. 

We identified pairs of residues present in all transferases that differed from those in the hydrolases. We focused on the pairs of residues in the top 10% of the central betweenness values. We detected pairs of residues differentially enriched in glycoside hydrolase clusters around F72 and F273 and selected F72/V86 from the first cluster and residues T274/M279 from the second cluster as mutagenic targets (for the enrichment values of these clusters see Appendix A). It is worth mentioning that position 279 is part of the fourth highly conserved sequence region in the GH13 family. We then investigated the role of the single and combined substitutions F72L, V86I, T274V, M279N in transglycosylation and hydrolysis reactions (Figure 4b, Appendix A). Although a decrease in hydrolytic activity was observed for many mutants tested in *TmGTase*, they produced a sharper reduction of transglycosidic activity in all of the variants, entirely impairing their activity in some of them (Table 2). As a result, the goal of increasing the H/T ratio was achieved. Of particular interest is the variant M279N, which, besides decreasing the transglycosidic activity by four-fold, it increased the desired hydrolytic activity by 25%, yielding a five-fold increment in the H/T ratio. It is worth mentioning that during the characterization of the variants, we detected a variant with a higher overall activity, which, besides the engineered F72L/T274V mutations, contained two unintended modifications: E77G/E226K. This variant does not change the H/T ratio because both reactions were favored by the extra substitutions, with an increase of about 40% overall activity relative to the wild-type protein (Figure 5b).

### 2.5. Molecular Dynamic Simulations

The role of protein dynamics in enzymatic catalysis is well recognized [64,65]. Thus, we carried out molecular dynamics simulations for some protein variants to explain the effect of mutations on the activities of the enzymes in structural terms. In the case of *TmAmyA,* the root mean square fluctuation (RMSF) difference between the triple mutant (K98P/D99A/H222Q) and wild-type protein was discreet (Figure 6). The α4 helix (catalytic domain), marked with residue 247, was more flexible in K98P/D99A/H222Q than in the wild-type *TmAmyA* (Appendix A and Figure 6a). This helix is near the loops containing the catalytic aspartate (nucleophile) and glutamate (acid-base). Contrarily, the K98P/D99A/H222Q mutant was rigidified at the loops and helices comprising residues 325, 387, and 415. 

The changes in flexibility were also studied for wild-type *TmGTase* against M279N single and T274V/M279N double mutants. An augmentation in the RMSF of the double mutant T274V/M279N was evident when compared to those in the wild type *TmGTase*, around residues 100, 107, 121 (B-domain), 131 (+2 subsite, B domain), 210 (loop α4β5), 222 (helix α5), 264 (helix α6), and 325 (loop α4β5, near subsite −3; Figure 6b and Appendix A). While RMS fluctuations for the single mutant M279N were similar to the double mutant around residues 100, 107, 121, and 131, they were increased relative to the other proteins near residue 222. Thus, the B domain, which is important for substrate specificity in other GH13 enzymes, is more mobile in both mutants [57]. 

The dihedral angles (χ_1–3_) of the catalytic acid-baseresidue of wild type and mutants from *TmAmyA* and *TmGTase* were analyzed, as elsewhere [47], and are shown in Appendix A. We observed that contrary to David et al., the χ_3_ angle showed a more diffuse distribution in the more hydrolytic variants of both enzymes (wild-type *TmAmyA* vs. K98P/D99A/H222Q, and wild-type *TmGTase* vs. M279N). Furthermore, we observed a larger conformational sampling of the catalytic acid-base in the more hydrolytic variant of each pair (Appendix A). As Lundemo et al. has pointed out, the residue chain’s mobility and orientation could be better described by the χ_1_ and χ_2_ angles. Moreover, when studying the cyclodextrin glucosyltransferase from *Bacillus stearothermophilus* NO2, a GH13 enzyme, Kong et al. defined a new angle for analyzing E253 in this bacillus CGTase—finding that it is more flexible in mutant L277M, which is less hydrolytic than the wild-type protein [66].

In *TmAmyA*, the χ_3_ angle mainly occupies two conformations in the triple mutant, while it seems not to have any preference in the wild-type enzyme (Appendix A). Further studies are required to elucidate the effect of this amino acid mobility and orientation on the reaction specificity. 

The change in structural dynamics modifies the pKa of E216 in *TmGTase.* PROPKA calculations [67] of this residue were performed on the structures corresponding to three different times in the simulation: 200, 300, and 400 ns, when the RMSD values plateau. The pKa of the catalytic acid-base residue was higher for both variants than for the wild-type *TmGTase*. For this parameter, the average was 3.0 ± 0.97 for the wild-type, and 6.1 ± 0.54, and 4.8 ± 0.43 for T274V/M279N and M279N, respectively. These results agree with the notion that hydrolysis requires a more basic residue than transglycosydation [41]. Although the pKa of the acid-base residue changes drastically during the reaction, this analysis suggests that the enzyme is tuned to increase its pKa. Despite considering only the free enzyme in the simulation, it is interesting to notice a shift in pKa—although the enzyme has not yet formed the covalently bound sugar-enzyme intermediate. Thus, these values have to be taken with care since they do not reflect the acid-base residue’s environment during the second step of the reaction. In the case of residue E258 of *TmAmyA,* the K98P/D99A/H222Q triple mutant has a pKa value similar to the wild type for its catalytic acid-base residue (around 4.8 for both). This result suggests a different mechanism for the change in reaction specificity, one exclusively affecting the hydrolysis reaction.

Additionally, the average distance between D278 and E216 was 1 Å closer in both mutants than in the wild-type enzyme *TmGTase* (Appendix A). As these two acid groups influence each other pKas, reducing the space increases the pKa of at least one of the participating amino acids, in order to avoid electrostatic repulsion. Consistently with PROPKA calculation, the average distance fluctuations between D310 and E258 are similar for *TmAmyA* proteins. These residues are closer in the wild-type *TmAmyA* than in its triple mutant, shifting to a further distance after 350 ns of simulation, and averaging the same for both variants (Appendix A). These results suggest a different dynamic for both glycosidases at the region close to D278, a residue that functions as a transition state stabilizer. Additionally, according to MD simulations, for the *TmGTase* T274V/M279N variant, D278 established a stable hydrogen bond with K324 that was not observed in other *TmGTase* variants (Appendix A). For *TmAmyA,* the equivalent residue to K324 was a Gly, unfit for forming a hydrogen bond with a corresponding catalytic aspartate.

## 3. Discussion

We implemented a methodology for identifying mutagenic target sites to modulate the transglycosylation/hydrolysis (T/H) ratio in two members of the GH13 family. This methodology selected target residues far from the active site (for this work between 11.1 and 22.2 Å away) that modified the reaction specificity (Figure 7a,b). It was interesting to note that it also could select residues close to the catalytic site, such as residue 279 of *TmCGTase,* which is next to the catalytic residue D278 (2.7 Å; Figure 7b). This reaction specificity modulation seemed to work in both directions; however, the more significant contribution was the reduction in the undesired reaction, and to a lesser extent, the increase in the desired one, in most cases. 

Conservation of residue-residue contacts can be used to recapitulate the clustering of some of the GH13 subfamilies (Figure 2), as well as the separation of the enzymes according to their reported specificity (Figure 1). In this classification, the number of transglycosidic residue-residue contacts (formed by two residues enriched in transferases) seemed to play a more prominent role in distinguishing between functions. Hydrolytic residue-residue contacts (constituted by two residues enriched in hydrolases) are low for some hydrolases. In contrast, transglycosidic residue-residue contacts were always abundant in transglycosidic enzymes and underrepresented in the hydrolytic ones. These results are consistent with previous works where patterns of contacts have been used to distinguish protein families [26] and add to their use in functional and evolutionary classification of subfamilies.

Our analysis detected many evolutionary relationships. It indicated that *TmGTase* and *TmAmyA* resulted from a gene duplication event after speciation, as these were the closest structures in the analysis, but the enzymes have different functions. The structural analysis also pointed in this direction. It was striking that *TmAmyA* was the nearest to the Firmicutes’ enzymes than to those of other Archaeas—a branch to which *T. maritima* belongs [68]. This advocates for a horizontal transfer between the two groups, resulting in a close relationship between AmyA’s GH13 subfamily 36 and Firmicutes’ subfamily 2, whose members include both bacterial and archaeal enzymes [69]. This relationship is also conveyed by the similarity of the B domains between the *TmGTase* structure (PDB ID: 1LWJ) and the other enzymes of the Firmicutes group. Nonetheless, domain B varied the most out of the three core domains of the GH13 family; residues 148–161 in 1LWJ could be readily aligned with 180–191 of the *Geobacillus thermoleovorans* CCB_US3_UF5 GTase (PDB ID: 4E2O), and many features were shared throughout the structures. It is noteworthy that this Geobacilli enzyme has not been assigned yet to a subfamily in the CAZy database. These results commend a more extended use of residue contacts in the classification of enzymes, complementary to sequence and structure analysis, as their analysis is more sensitive than sequence analysis and can be automated.

Results from directed evolution, especially in CGTases, have emphasized the relevance that residues at subsites +1 and +2 have on determining reaction specificity [50,51]. In these works, however, as the authors recognize, a completely random mutagenic scheme mainly allows the exploration of single mutants. The opportunity to guide the mutagenic process with structural or mechanistic information would accelerate the evolution of reaction specificity in these proteins. The successful combination of mutations has required several rounds of mutagenesis and screening in order to find them. Still, most of the work reported on alpha-amylases explores only the active site. The role that dynamics play in stabilizing the transition state of reactions is much subtler but no less important. This contribution is encoded in the protein sequence, and decoding it through the analysis of contact conservation patterns expands the exploration to positions beyond the active site, in order to influence reaction specificity. Thus, our analysis of contact conservation patterns detects residues that have already been explored and that change reaction specificity. For example, it predicts the substitution A230V in CGTase from *B.circulans* [51], and suggests the substitution of other residues in contact with A230 that have not yet been investigated to improve the hydrolysis reaction even more. 

As part of the results obtained with the implemented algorithm, we evaluated the rationality of other reported mutations in glycoside hydrolase enzymes. The mutation A289F, that introduces tranglycosylation activity in *B. stearothermophillus* α-amylase, could have been predicted by a change in enrichment factors from −0.03 to +0.1 [45]. It also occurs with the mutation V286F in *B. licheniformis* α-amylase (BLA) [44], with a similar change in enrichment factor values. Interesting mutants on which our group are now working are V286F/T329M (*B. licheniformis*) and A289F/T335M (*B. stearothermophilus*), which include a new mutation suggested by the enrichment factor change from −0.3 (T) to +0.7 (M). Although a mutation in H222Q in *TmAmyA* would not have been chosen, the substitution of histidine at that position (enrichment factor −0.05) to leucine (enrichment factor: 0.25) is indicated by the analysis of enrichment factors to increase transglycosidation.

As for the mutations characterized in this work, the equivalent residue to *TmGTase* T274 in *Aspergillus oryzae* α-amylase was occupied by V293, rendering a functional α-amylase (Appendix A). This loss of function might have been provoked by a need to remodel the network to which T274 belongs. (Appendix A), emphasizing again the necessity of a methodology for identifying groups of residues that interact together. In conjunction with residue 274, position 279 is involved in a hydrogen bond network comprising D314, R281, F311, T274, S275, N276, K244, F273, T274, M279, and S280 (Appendix A).

The mutations identified in this work seemed to be connected to the catalytic site, probably influencing the catalytic site in addition to the dynamic changes described earlier. One of such elements was linked to the calcium ion to which TmAmyA D99 was bound, which might reflect once again the importance of metal ions in α-amylase structure and function [36,70] (Appendix A). As observed in this work for the *TmGTase* variants, mutations modifying the transglycosidation/hydrolysis (T/H) ratio change the dynamics of the loops surrounding the protein’s active center, including the B domain. This behavior agrees with the GH51 retaining α-l-arabinofuranosidase from *Thermobacillus xylanilyticus* [71] and the almond β-glucosidase, in which the movements of four strategically located loops act as a lid for the active center, controlling the catalytic activity [72]. 

Additionally, in the GH13 family, a chimeric amylosucrase from *Deinococcus geothermalis* (DGAS) and *Neisseria polysaccharea* (NPAS) had a differential fluctuation in loops 4, 7, and 8 between both variants, associated with T/H changes [73]. The mutation A226N in DGAS also modified the T/H ratio from 0.59 (wild type) to 0.9 and diminished the flexibility in loops 2, 3, 4, 7, and 8 [74]. Additionally, it was reported for the *T. kodakarensis* glycogen branching enzyme (family GH57) that a loop extending from G227 to P248 contained a Tyr (residue 233) whose modification changed the branching/hydrolysis ratio from 41 (wild-type) to 16.2 without affecting branching activity [75]. 

For the helix extending from residue 221 to 231, we observed a change in mobility and T/H ratio for *TmGTase* M279N. Changes in specificity have been observed for mutations in the equivalent helix. In *Bacillus stearothermophilus* amylase (BStA; from I270–K279), the mutation I277F increases the specific activity of this hydrolase [76]. This result might also explain why F72L/E77G/E226K/T274V has a higher hydrolytic activity than the F72L/T274V variant as residue E226 interacts with two charged residues (D225, R22), producing a kink in the helix. Substituting residue E226 for a positively charged residue (E226K) might change the inclination and dynamic of this helix, thus affecting the catalytic acid-base. Together, these results suggest the importance of the dynamic reconfiguration of the helix comprising residues 221–231 in hydrolytic activity. 

The comparison of residue contacts between groups might find applications outside of function, as suggested by a work in which the stability of a GH13 enzyme was modified by mutagenesis at hotspots identified by direct coupling analysis [77], creating and removing hydrogen bonds between residues. It also could identify sites where fluorescent probes can be inserted while minimizing functional impairment. This methodology finds its parallel in some coevolution analysis methodologies that have been reported [29,30]. Still, it required less data to identify sites crucial for function, as contacts were directly observed from the structure while the coevolution analysis inferred them. Both perspectives highlight the importance of analyzing residue contacts when a function is studied, even away from the catalytic site. Previous works aiming to classify enzymes of unknown function and predict and verify 3D models have also suggested the importance of residue contacts [25,26], where contact map prediction has played an important role.

The method presented in this work is limited by the availability of 3D structures, which are usually less abundant than the sequences employed in coevolutionary studies. Both strategies benefit by including many proteins in the study. Still, as mentioned earlier, the contact map analysis requires fewer proteins, as contacts are obtained directly from structures. In contrast, coevolution studies infer this information via correlation analysis. This limitation could be overcome by including sequences in the analysis aligned to the closest structure, but it should be considered that the number of enzymes experimentally characterized as 1,4-α-glycoside hydrolases is much higher than those of the 1,4-α-glycosyltransferases (only 31), at least as reported in the CAZy database. 

Another limitation is the incomplete information available on the reaction specificity of each enzyme in the glycoside hydrolase family. This analysis could also be enriched if characterization of both functions was available for all enzymes. This characterization would also determine if there is a functional meaning to the grouping of GH13 enzymes based on their hydrolytic or transglycosidic contacts. This methodology should also be extended to include more simultaneous mutation sites than the maximum of four that was evaluated in this work. The residues explored were in contact with others forming a more complex network. Therefore, exploring mutants comprising a complete network of interacting residues should give a more noticeable change in reaction specificity. Despite these limitations, contact maps could be an additional auxiliary in the quest to predict critical functional residues as they are already being used to predict structures.

Our methodology aimed to create a parameter to guide the modulation between transglycosidation and hydrolysis specificity. As Healp and Blouin pointed out [78] in their study of the evolvability of the GH13 family stability, a quantitative prediction of specificity requires catalytic information with a single substrate for all the members of the family under analysis. In contrast, the GH13 family has been characterized with a great range of substrates for both the hydrolytic and transglycosidic activities, and in many cases, just for their most prominent activity. Even when the most common substrate, starch, has been used, variations in 1,6-ramification might preclude this characterization. We surmounted this obstacle by performing a differential analysis of enzymes with a distinct specificity, thus creating a parameter (the enrichment factor) that offers a qualitative guide to transform reaction specificity that might require more experimental data in order to become quantitative. Nevertheless, this guide might reduce the sequence space that needs to be explored in order to achieve a specificity change, both in rational and directed evolution studies.

The development of computational approaches to resolve biological problems is a growing research area [79,80]. Many aspects of the protein structure–function relationship are of particular interest in engineering enzymes for biocatalysts. Additionally, artificial intelligence could be an approach for improving and complementing other strategies that are already employed for mining functional data. This tool could help overcome the effects of epistasis, which occludes the selection of combined mutations, to improve the desired function. Recently, Timonina et al. [81] reported a method based on artificial intelligence, named Zebra3D, which centers its analysis on specific structure regions (SSRs) of the protein family after aligning their 3D structures. It classifies proteins into subfamilies, with distinct structural elements for each enzyme associated with substrate specificity in human aldose reductase and catalytic activities for α/β-hydrolases. Analyses such as Zebra3D would benefit by including the study of residue contacts. Residue contact analysis not only classifies enzymes functionally, but also identifies distant interacting regions—especially when networks of contacts are considered. The investigation of more extensive networks is underway.

## 4. Materials and Methods

### 4.1. Bioinformatic Analysis

#### 4.1.1. Analysis 1

CMView 1.0 [55] was used to create the contact map (cut-off distance 5 Å, all atoms) of 14 members of the CAZy family GH13 (Dataset 1, Appendix A) and were aligned against the contact map of the *Thermotoga maritima* 1,4-α-glucanotransferase *TmGTase* (PDB: 1LWJ). All selected structures were bound to a transition state analog, 12 were hydrolases and 4 were transferases. We wrote an R program to obtain the *TmGTase* residue-residue contacts shared with the rest of the structures and their corresponding amino acids. The algorithm also obtained the frequency of each amino acid at each position for all contacts ($f_{aa}^{ij}$) within a group (hydrolase,$\;f_{aa}^{ij,H}$, or transferase, $f_{aa}^{ij,T}$) and an enrichment factor ($\Delta f_{aa}^{ij}$) defined as:(1)$$\Delta f_{aa}^{ij}=f_{aa}^{ij,T}-f_{aa}^{ij,H}$$

We qualified the 14 structures used, and an additional set of 14 enzymes (twelve hydrolases and two transferases with structures without any ligand bound; Dataset 2, Appendix A) as transferases or hydrolases based on the enrichment factor scored by both residues at each contact. To do this, the residue contacts were compared with those of *TmGTase* (PDB ID: 1LWJ). The amino acids in all residue contacts shared with 1LWJ were qualified as hydrolytic if $\Delta f_{aa}^{ij}<0$ (the amino acid was more frequent in hydrolases) and transglycosidic if $\Delta f_{aa}^{ij}>0$ (the frequency of the amino acid was higher in the transferases). After qualifying all the amino acids in the residue contacts, contacts formed with a pair of residues deemed transglycosidic were designated transglycosidic; those formed by two hydrolytic residues, as hydrolytic.

During the development of this work, new members of family GH13 were included in the CAZy database, so that we created Dataset 4, which consists of 71 members—40 classified as hydrolases and 31 as transferases. Dataset 4 was used to calculate the enrichment factors, and then to classify hydrolases or transglycosidases based on their contacts, which were based on the enrichment factors.

Dataset 3 was created to compare contact conservation as a classification criterion with other probed methods such as sequence alignment [82] or structure conservation [83]. This data set contained proteins from Dataset 1 and Dataset 2 and eleven additional proteins classified as members of the GH13 family in CAZy—a GH97 enzyme (PDB: 2ZQ0), and a GH31 enzyme (PDB:3W37)—all acting on 1,4-α-glycosidic bonds either as a hydrolase or transferase (Appendix A). All of the sequences were aligned with ClustalW [82]. The structures were aligned using the DALI server [83]. In the case of the contact maps, all the contacts for every protein were identified. The fraction of enzymes sharing each contact was calculated (contact conservation score). All enzymes were used as a reference to calculate the conservation of all the contacts, and then compared against those of every other protein in the dataset. The contact conservation score sets of all the enzymes were plotted against each other.

#### 4.1.2. Analysis 2

Enrichment factors of pairs of amino acids were calculated from Dataset 1. The contact maps and alignments obtained from CMView were analyzed using 6 Å as a cut-off for all the atoms. The R program used for analysis calculated the frequency of all amino acid pairs for all contacts ($f_{aap}^{i}$) and an enrichment factor for the pair:(2)$$\Delta f_{aap}^{i}=f_{aap}^{i,T}-f_{aap}^{i,H}$$

The contact map of *TmGTase* was transformed into a 3D network, whose nodes were the α-carbon of each residue and the edges of the residue contacts. For this network, the central betweenness of its nodes was calculated. The betweenness centrality (BC) of a node *x* that is part of a network V was calculated through the following equation:(3)$$BC_{x}=\overset{N}{\underset{u,\;v\in V}{\sum }}\frac{\sigma_{u,v}\left(x\right)}{\sigma_{u,v}}$$
where *σ**_u_*_,*v*_ is the number of paths between the nodes *u* and *v*, and *σ**_u_*_,*v*_(*x*) is the number of times these paths contain the node *x* [62].

#### 4.1.3. Analysis 3

Enrichment factors for all the contacts in the homology model of the *Thermotoga maritima* α-amylase AmyA (*TmAmyA_2*) were calculated. *TmAmyA_2* was compared against 71 structures of GH13 enzymes (31 transferases and 40 hydrolases, Dataset 4, Appendix A). Contact maps and alignments were produced with CMView using a cut-off distance of 5 Å between all atoms.

#### 4.1.4. Selection of Mutation Sites

The results from Analysis 1 were used to identify positions in the homology model *TmAmyA_1*. Mutation sites were selected based on their conservation score, enrichment factor and their distance from the catalytic site. The conservation score of the sites considered was above 10 (out of 14). The occupying amino acid should have scored an enrichment factor below or equal to 0, with the alternative to be substituted with an amino acid having an enrichment factor above 0.2. Residues not in contact with the transition state analog were preferred.

Additionally, results from Analysis 2 were used to identify positions in *TmGTase* as targets for mutagenesis. Mutation sites were selected based on their conservation score (above 10), enrichment factor (top and lowest 1%), and their central betweenness score (top 20%). 

The code for the enrichment factor calculation is available at: https://github.com/rhodbacter/ContactMaps-Catalysis/ (accessed on 26 July 2021).

#### 4.1.5. Molecular Dynamic Simulations

The molecular Dynamic (MD) simulation and analysis were accomplished using GROMACS (GROningen MAchine for Chemical Simulation) version 2020.4 [84], as reported [85], with slight modifications. The atom coordinates for each protein were obtained from the Protein Data Bank [86]. All modification or visualization of PDB files was carried out with UCSF Chimera [87] or Pymol (Schrödinger, Inc., New York, NY, USA). Incomplete structures were completed with Modeller [88]. Hydratation was performed using “Simple Point Charge Extended” (SPCE) in a cubic box of 1 nm × 1 nm × 1 nm. The forcefield was OPLS-AA/L (optimized potentials for liquid simulations). The system was neutralized with Na^+^ and Cl^−^ ions. Later, minimization by steepest descent minimization was performed until the energy decreased to <1000 kJ/mol/nm. Equilibration was run using a leap-frog integrator and with a modified Berendsen thermostat for 1000 ps, with a step of 500,000—first for a NVT ensemble, followed by one with an NPT ensemble. Finally, the simulations were performed for 500 ns with an NPT ensemble using a leap-frog integrator and a modified Berendsen thermostat. Analysis of results were done with GROMACS tools, and visualization was accomplished with Xmgrace [89]. The pKas of aspartic and glutamic acids were calculated with PROPKA 3.4.0 [67].

#### 4.1.6. Homology Models

All homology models were created using the Swiss Model [90,91,92,93,94,95,96,97,98] with the best matching template. The quality of the models was assured through their QMEAN values (Appendix A). Additionally, the validation of the model obtained for *TmAmyA* was done with the structure assessment tools (QMEAN, Ramachandran favored, and MolProbity Score) from the Swiss-Model server [58], VERIFY 3D [99,100] and ProSA-web [101,102].

### 4.2. Experimental

All reagents were purchased from Sigma-Aldrich (St Louis, MO, USA) unless stated otherwise. All enzymes were purchased from New England Biolabs (Ipswich, MA, USA) unless otherwise stated.

#### 4.2.1. Construction of the TmGTase Gene Vector

The Tm0364 (KEGG ID) gene was amplified from a *T. maritima* cDNA library by PCR reaction using *pfu* DNA polymerase. The PCR product was purified using agarose gel electrophoresis and Roche PCR purification kit (Roche Diagnostics GmbH, Mannheim, Germany). Vector pET22a and the PCR product were digested using the restriction enzymes *Nde*I and *Xho*I. The resulting fragments were purified through agarose gel electrophoresis and the Roche PCR purification kit (Roche Diagnostics GmbH, Mannheim, Germany) and finally ligated using *T4* DNA ligase (ThermoFisher, Waltham, MA, USA). The ligation product was used to transform the genes into electrocompetent *Escherichia coli* MC1061 cells. The *TmAMyA* gene was cloned in the same way, and it has been previously reported [43].

#### 4.2.2. Construction of TmAmyA and TmGTase Variants

Mutations were constructed either through the Mega Primer method (using *pfu* DNA polymerase) [103] or Quick Change (with *Dpn*I and ThermoFisher Scientific *phusion* high fidelity enzyme) using oligonucleotides (Appendix A) purchased from Unidad de Síntesis y Secuenciación de DNA of Instituto de Biotecnología, UNAM.

#### 4.2.3. TmAmyA and TmGTase Variants Expression

All the pET22a plasmids containing the genes were transformed into calcium competent *E. coli* K12 ER2738 cells. Induction was performed with 0.5 mM IPTG (*TmAmyA* variants) or 0.1 mM IPTG (*TmGTase* variants). In all cases, purification was performed over two steps after sonication cell lysis: (1) heating at 70 °C for 1 h, and (2) affinity chromatography of the heated supernatant with a Ni-NTA agarose column (ThermoFisher Scientific, Waltham, MA, USA) following the supplier’s protocol. Protein concentration was determined based on the Bradford method [104] using a Pierce™ Coomassie Protein Assay Kit, following the instructions of the manufacturer.

#### 4.2.4. Characterization of TmAmyA Variants

An alcoholysis reaction was started by adding 20 U (µeq of dextrose produced per min) of enzyme to 1 mL of 100 mg/mL starch in 10% butanol, 50 mM Tris, 150 mM NaCl, and 2 mM CaCl_2_ at pH 7.0, and measured after 12 h of reaction at 85 °C. Hydrolysis and transglycosylation products were measured as reducing sugars by DNS reagent [105] and as the formation of butyl glycoside by HPLC after digestion of the reaction products with glucoamylase (Sigma-Aldrich, St. Louis, MO, USA), respectively. HPLC analysis was performed in a Waters-Millipore 510 HPLC system equipped with an automatic sampler (model 717 Plus, Waters Corp., Milford, MA, USA), a refractive-index detector (Waters 410, Waters Corp., Milford, MA, USA) and a Hypersil GOLD™Amino column (Thermo Scientific, Wilford, UK), using acetonitrile:water (80:20) as the mobile phase at a flow rate of 1.0 mL/min. 

#### 4.2.5. Characterization of TmGTase Variants

Tranglycosylation activity was measured in the reactions, started by the addition of 12 µg of enzyme to 1 mL of 5 mg/mL maltoheptaose in 50 mM Tris, 150 mM NaCl, 2 mM CaCl_2_ pH 7.0 and kept up at 70 °C. Finally, quantification of sugar complex with iodine reagent [106,107,108,109] after 1, 2, 5, 10, 15, 30, 60 and 90 min was measured at 580 nm and reported as the starch equivalents produced.

In other independent experiment, hydrolytic reactions were started by the addition of 60 µg of enzyme to 1 mL of 10 mg/mL starch in 50 mM Tris, 150 mM NaCl, 2 mM CaCl_2_ pH 7.0. Hydrolysis at 70 °C was measured by the reducing sugars produced using DNS reagent [105] after 6, 12, 18 and 24 h.

## 5. Conclusions

We present a method based on contact maps that reveals structural features important to function, even if they are not part of the protein sequence—such as the ions identified in this study. The study results showcase the ability to analyze residue contacts in order to bring new insights to our understanding of protein function beyond than the catalytic site. We believe this process can be extended beyond protein function to other properties such as thermal stability, tolerance to salinity, pH, or pressure. As in other studies employing contacts between residues, we observed that residue contacts are a tool that permits the identification of protein families that can be complemented by employing enrichment factors, as presented in this work, allowing the identification of subgroups. This could be used as a tool complementary to analysis through coevolution of residues. Both techniques overcome their limitations by interchanging information and using such as artificial intelligence that are more readily available every day. 

## Figures and Tables

**Figure 1 molecules-26-06586-f001:**
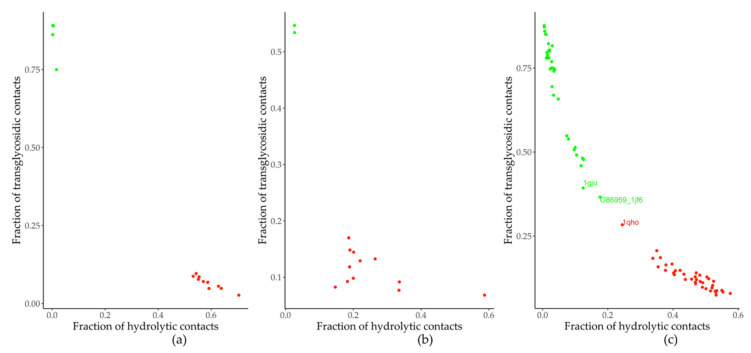
Function classification of enzymes in the GH13 family based on the amino acid preference in their residue-residue contacts. (**a**) Dataset 1 (training group). (**b**) Dataset 2 (validation group). (**c**) Dataset 4 (updated dataset). In both cases, enzymes defined as transglycosidic or hydrolytic are shown with green and red dots, respectively.

**Figure 2 molecules-26-06586-f002:**
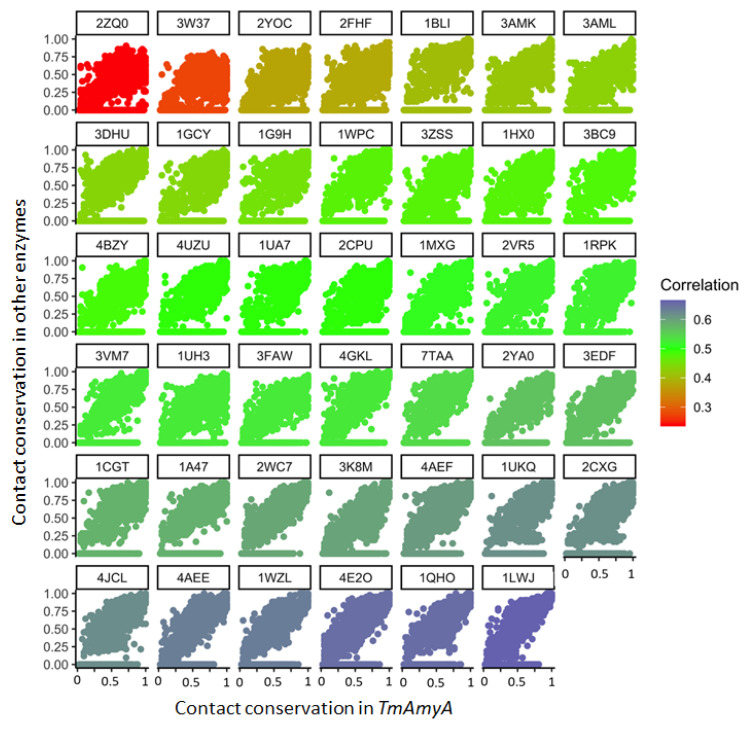
Correlation of the residue-residue contact conservation between all enzymes and those of TmAmya, one of the enzymes modified in this study. The preservation of each contact in a protein is plotted against its conservation in TmAmyA. This parameter seems to correlate with phylogenetic relationships. A poor correlation is evident for the enzymes not belonging to the GH13 family (e.g., PDB ID: 2ZQ0, family GH97, and enzyme with the PDB ID: 3W37, family GH31).

**Figure 3 molecules-26-06586-f003:**
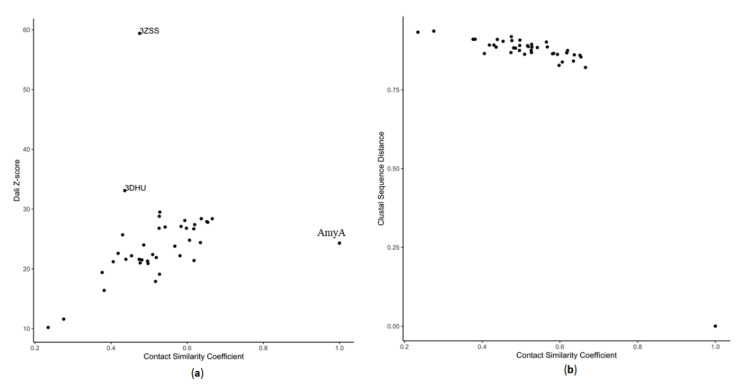
Parameters of similarity distance between evaluated enzymes and TmAmyA based on their correlation between contact conservation (R-squared) and (**a**) structure (Dali) and (**b**) sequence (Clustal) alignments.

**Figure 4 molecules-26-06586-f004:**
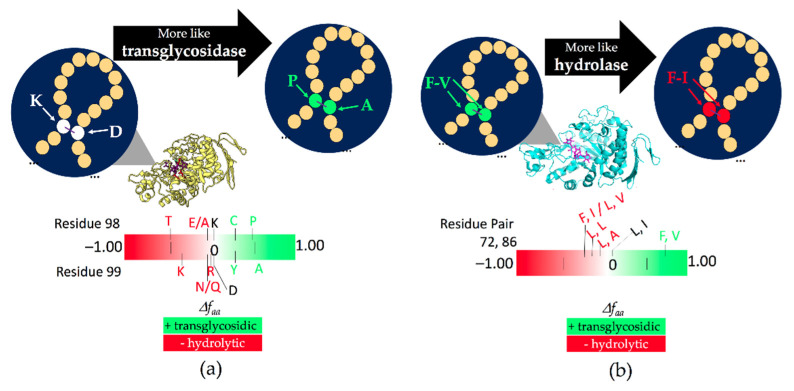
Schematic representation of substitutions in glycosidases based on enrichment factors. (**a**) K98P/D99A in TmAmyA (**b**) F72L/V86I for TmGTase. The residues more frequently found in hydrolases and transglycosylases are shown in the lower part of the figure for each pair in red and green, respectively.

**Figure 5 molecules-26-06586-f005:**
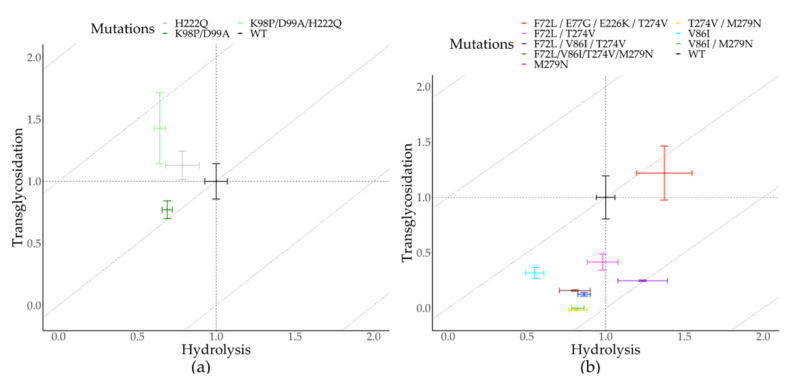
Transglycosylation/Hydrolysis (T/H) ratio of glycosidases. (**a**) TmAmyA and its variants; (**b**) TmGTase and its variants. Transglycosidation and hydrolysis values were normalized to each wild-type enzyme, which has the coordinates (1,1). The dashed lines indicate equivalent T/H ratios. The central diagonal corresponds to the wild-type enzyme. Enzymes with a better T/H ratio are over the central diagonal line, and enzymes with a better H/T ratio are under the central diagonal line.

**Figure 6 molecules-26-06586-f006:**
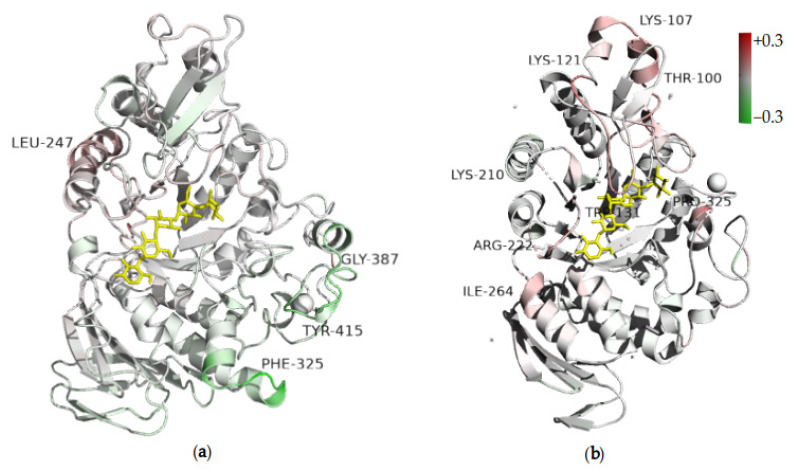
RMSF difference (in nanometers) along the structure with high and low transglycosidic variants of glycosidases. (**a**) TmAmyA K98P/D99A/H222Q triple mutant (less hydrolytic than the wild-type enzyme). (**b**) wild-type TmGTase (high transglycosidic) vs. T274V/M279N. The active center is delimited by acarbose (yellow stick) and is located away from the zone with the modified fluctuations. No differences in RMSF are shown in white, while positive or negative RMSF changes are red and green, respectively.

**Figure 7 molecules-26-06586-f007:**
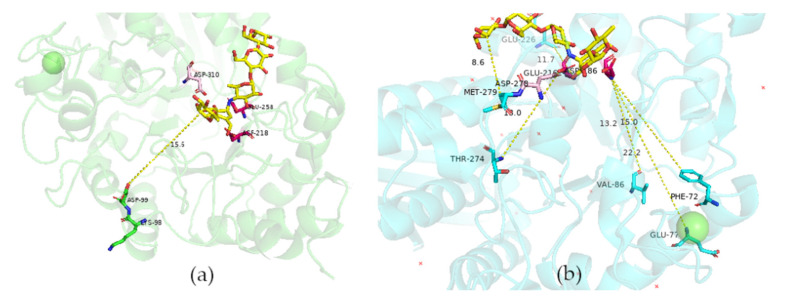
Distances in Angstrom between the mutation sites (green and cyan) and the catalytic residues (red) (**a**) TmAmyA. (**b**) TmGTase. The active center is delimited by the competitive inhibitor acarbose (yellow). In pink is shown the D310 residue that acts as a transition state stabilizer.

**Table 1 molecules-26-06586-t001:** *TmAmyA* variants production after 12 h of reaction.

*TmAmyA* Variant	Hydrolysis (mEq Dextrose/µg Protein × 10^−2^)	Transglycosidation (mEq Butyl Glucoside/µg Protein × 10^−4^)	Transglycosidation/Hydrolysis (T/H) Ratio × 10^−2^
Wild type	2.8 ± 0.2	7 ± 1	2.5 ± 0.6
K98A/D99P	1.93 ± 0.09	5.4 ± 0.5	2.8 ± 0.4
H222Q	2.2 ± 0.3	7.9 ± 0.8	3.6 ± 0.4
H222Q/K98A/D99P	1.8 ± 0.1	10 ± 2	5.6 ± 0.4

**Table 2 molecules-26-06586-t002:** *TmGTase* variants specific activity.

*TmGTase* Variant	Hydrolytic Activity (×10^−5^ mg Starch/µg Protein/min)	Transglycosidic Activity (×10^−3^ mg Starch/µg Protein/min)	Hydrolysis/Transglycosidation (H/T) Ratio (×10^−2^)
Wild type	5.1 ± 0.3	4.0 ± 0.8	1.3 ± 0.3
V86I	2.9 ± 0.3	1.3 ± 0.2	2.2 ± 0.6
M279N	6.3 ± 0.8	1.01 ± 0.03	6 ± 1
V86I/M279N	4.2 ± 0.2	ND	NA
T274V/M279N	4.2 ± 0.3	ND	NA
F72L/V86I/T274V	4.4 ± 0.2	0.51 ± 0.07	9 ± 2
F72L/V86I/T274V/M297N	4.1 ± 0.5	0.65 ± 0.03	6 ± 1
F72L/E77G/E226K/T274V	7.0 ± 0.9	5 ± 1	1.4 ± 0.5
F72L/T274V	4.9 ± 0.5	1.7 ± 0.2	2.9 ± 0.6

ND: non-detectable activity; NA: not available.

## Data Availability

Not applicable.

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
