# Peer review of "Modulating Glycoside Hydrolase Activity between Hydrolysis and Transfer Reactions Using an Evolutionary Approach"

_molecules, 2021, doi:10.3390/molecules26216586_

Round 1

Reviewer 1 Report

See attached file.

Author Response

Manuscript ID: Molecules-1412390 entitled, "Modulating glycoside-hydrolases activity between hydrolysis and transfer reactions using an evolutionary approach" by Rodrigo Arreola, Alexey Llopiz, Leticia Olvera, and Gloria Saab-Rincón

Response to Reviewer 1

General remarks The authors describe the modification of two GH13 enzymes with different reaction specificities (hydrolase (H) and transferase (T)), using a computational approach to identify residues outside their active site that may affect the original reaction specificity towards the other. The computational approach is based on conservation of coevolving residues within GH13 enzymes. Mutants of the two enzymes were then constructed and tested for their T/H ratio; it was found that rather than increasing the desired specificity, the undesired specificity decreased, still leading to the desired improved T/H ratio. MD simulations were done to explain the observed effects; flexibility of residues and regions, as well as hydrogen bonding networks seem to play a role herein. This is an interesting approach that might be applicable to other enzyme families/specificities. The interpretation of the results depend in my opinion on reliable structures. The two enzymes used in this study are a crystal structure and a homology model, but it is unclear how reliable the latter is. The manuscript is in general fairly easy to read, but in some parts rather lengthy. Especially the Discussion is too long in my opinion; it might also benefit from adding section headings. Perhaps it would also help limiting the number of figures in the main text (now 14) and move some of them to the supplementary material.      

Specific points

  • The first part of the Results section describes a few datasets (1 to 4); it would be helpful to the reader to clearly define/describe each dataset and its aim with regard to the analyses.

Response: Dataset 1 only contained proteins whose structure bound to acarbose was reported. since this was the training dataset we wanted to resemble more the active conformation. Dataset 2, contained fourteen structures of proteins no-bound to acarbose, this set was used to evaluate the method. Dataset 3 included dataset 1 and 2 and 14 new proteins from CAZy, besides two members of other glycoside-hydrolases families. This broader set was used to analyze the correlation of the contact-conservation among members of the family GH13 and outsiders.

Finally Dataset 4 was an updated from dataset, including new glycoside-hydrolases added in CAZy during the development of this work.
The explanation of the selection of each Dataset has now been included in the corresponding sections.    

  • Line 128: “…design of enzymes…” The first part of Figure 2 legend is hard to understand, as is in fact the figure in itself. Is it possible to more carefully describe the concepts and what is represented?

Response: We have tried to clarify Figure 2 legend and also added a description of the concept of contact conservation correlation (lines 199-209)

  • Line 250-251: What is meant with “clustal sequence distance”? Figure 4: The green text on a blue background is very difficult to read; please change colour.

Response: The mean is a pairwise sequence distance. Thi term has been replaced in the text(line 220). And the color has been changed to contrast better the green colour in the background.

  • Line 392-394: Here, the authors chose a different analysis than in the previous case. It is not completely clear to me why this choice was made. Wouldn’t it be more fair to use the same type of analysis?

Response:The rationale was that while for hydrolases, choosing the more abundant residues in a pair could give a pair represented in de Dataset 1, the lower representation of transferases in Dataset 1, could suggest a non-compatible pair. This explanation has been added (lines 301-304). Also pairs are more importante in transferases than in hydrolases as shown in Figure 1.

  • Line 395-397: The concept of the “betweenness centrality” parameter is not very clear to me, and why this was needed to restrict the search of mutation sites. A somewhat more detailed explanation would be helpful to understand the concept and reasoning.

Response: Betweenness centrality (BC) of a node or vertex denoted as x that is part of a network V is calculated through following equation:

where ?u,v is the number of paths between the nodes u and v, and ?u,v(x) is the times that these paths contain the node x. In a biological system is a measure of the importance of the node in the network, in this case a residue contact network. The explanation of this parameter was added in the manuscript. The concept was explained in the Results section and also added to Methods section.

  • Line 470-543: The description of the angles and their distribution  for the different mutants is rather lengthy. In my opinion, the corresponding graphs (figures S3 and S5) are ‘easier’ to interpret, and in the text here a short overall description would be sufficient.

Response: The distribution of the dihedral angles was presented in a summarized form.

  • Line 641 and Figure 12: the distance as indicated in Figure 12 is 7.4 Å; how can these residues interact via hydrogen bond? Is that a result of dynamics in the MD simulation?

Response: The residues are at a distance of 7.4 Å in the crystallographic structure. The hydrogen bond was detected as a result of molecular dynamic simulation. Also, the text and the figure’s footprint were corrected.

  • Line 648-649: “D278, which plays no direct role in the catalysis” - one might agree, but although it is not involved in transferring atoms during the reaction, it is considered part of the catalytic triad. Perhaps mention that it is the transition state stabilizing residue? Response: It has been clarified that it is part of the catalytic triad.
  • Lines 650-651: Can the authors speculate as to why the undesired reaction was reduced rather than an increase of the desired reaction? 2

Response: Reduction of an activity can be achieved in many different ways, like misaligning the acid-base relative to the nucleophile to be activated, changing the local pH around a catalytic residue, producing steric hindrance for the incoming acceptor, etc. Gaining activity implies fulfilling all the conditions for the catalysis. In the case of reaction specificity, mutations may produce a reduction of the effective concentration of only one of the acceptors, favoring the reaction with the other.  The difference between the number of transglycosidic contacts in hydrolases and transglycosidases suggests that more than one mutation will be required to consolidate the modification of reaction specificity. Thus, a modification of networks might be needed to achieve this.  As it is observed in the molecular dynamics for variants TmGTase M279N and T274V/M279N although in both variants a pKa shift of the catalytic acid/base towards an increase in hydrolysis, in mutant T274V/M279N it is observed that K324 forms an ion-ion interaction with D278 (transition state stabilizer), sequestering K324 to bind any incoming substrate and thus diminishing the overall enzyme activity.

  • Lines 835-857: It would be helpful to add a conclusion about these results at the end of this paragraph.

Response: This section has been considerably reduced but the conclusion was added and it is now in lines 486-490.

  • Lines 1009-1012 / 1025-1026: “…exploring mutants implying a complete network of interacting residues…” Can the authors comment on how this would have to be achieved? For example, there are studies where an enzyme has been mutated at every position, to every other residue type (mutatability landscape). Is this feasible for the GH13 enzymes?

Response: It is possible to explore each position at once with all or a limited residue type. But a change of reaction specificity may require simultaneous mutations at several residues. Exploration of three positions simultaneously requires the screening of over 32 000 clones. The proposed method is intended to limit the variability generation to reduce screening effort. At this point, we only explored specific pairs. The work underway makes multiple mutations at sites that form an interaction network. Residues investigated in this combinatorial mutagenesis are those with the highest ∆faaij values and the original residue at each position in the network. This is now emphasized in the Discussion section.

  • Also, is that what is meant in the last sentence of the Discussion? The TmGTase structure investigated is an experimental (crystal) structure; however for TmAmyA a homology model was constructed. How reliable is the latter, and what are the values of the parameters regarding reliability? It is only stated that SwissModel was used with the best matching template (which protein?). In addition, I think it is important to state this somewhere in the main text (both in Results and Discussion).

Response: The protein structure used as template is from the alpha-amylase from Thermotoga petrophila (PDB ID 5M99, resolution 1.96 Å). There are 30 extra residues at the N-terminus of TmAmyA than the crystallized amylase from Thermotoga petrophila (TmAmyA numbers are in according to Liebl et al., J. Bacteriol. 1997, 179, 941–948). The 504 remaining residues have 98.4 % sequence identity, showing only six substitutions. The N-term could not be modeled either by Swiss Model or by AlphaFold2. The following tools were used for the structural model validation: (1) “Structure Assessment” from Swiss-model (https://swissmodel.expasy.org/assess), (2) ProSA (Nucleic Acids Research 35, W407-W410) and (3) VERIFY 3D (https://saves.mbi.ucla.edu/). For ProSA-web Z-Score of -9.95 was obtained. This score for the query input structure was within the viewed range for experimentally-determined structures with similar size. Through VERIFY3D analysis, it was found that 99.21% of the residues have averaged 3D-1D score >= 0.2. This information was stated in the following sections: Results, Discussion, and Methods.

  • Lines 1146-1148: In this study the authors focus on hydrolysis versus transglycosylation as one aspect of reaction specificity. As the authors rightfully state, the methodology may be useful to study other properties (stability, saline tolerance, pH etc). Other glycosyl hydrolases also have differences in other aspects such as linkage specificity (e.g. α-1,4 / α-1,6), or product size specificity. Would the methods applied in this study be useful in modifying those aspects?

Response: Effectively, the method could be applied to any other property as far as enough proteins are characterized for the distinctive property to feed the training set.

  • Figure S1: might be added to Figure 1 as a third panel.

Response: The figure S1 hase been integrated in Figure 1.

Specific minor points

Line 46-47: I would suggest: “…residues outside the catalytic site and their contacts can play essential roles…”

Response: Thank you this suggestion was accepted.

Line 66: “A more direct attempt to exploit…”

Response: Thank you this modification was made.

Line 85-86: “…inherent to the protein…”

Response: Thank you this was corrected

Line 90: “…since proton removal…” (remove “a”)

Response: Thank you this was corrected

Line 102: “Multiple reports describe the modification of amino acid residues…”

Response: Thank you this was corrected

Line 110: the definition of MSA (multiple sequence alignment) should be given here, and can then be removed in line 118.

Response: Thank you this definition was moved above, as suggested.

Line 115-116: “…in many cases, the coevolution of residues is hard to detect…”

Response:Thank you. This has been edited.

Line 238: “This is an expected result since these enzymes do not belong…”

Response:Thank you. This was rewritten as suggested

Line 448-449: “…with a higher overall activity which, besides the engineered F72L/T274V mutations,…’

Response:Thank you. This was rewritten as suggested

Line 456 (Figure 5 legend): “Transglycosylation”

Response:Spelling was corrected.

Line 466: The definition of RMSF should be given.

T Response:hank you. The definition of the initial were added

Line 470-520: Instead of “As far as…” use “Regarding…” Figure 7: One might add the labels TmGTase (blue bars) and TmAmyA (orange bars) within the graph.

Response: “As far as” was replaced by Regarding

The data in Figure 7 is presented in the text, so that following the suggestion of another reviewer regarding the reduction of figures, this figure was removed.

Line 631: It would be helpful to include that D278 is the transition state stabilizer, and E216 is the acid/base.

Response:In Line 366, where we talk about the acid-base residue, we have included “E216”, to clarify that this is the residue that carries out this function. And in line 402, the role of D278 as transition state stabilizer is pointed out

Line 838-839: “…shows more flexibility for K98P/D99A/H222Q in the α4-helix in which L247 is located.”

Response: It was rephrased as suggested.

Line 974: “This result also agrees with the results obtained for…”

Response: This was corrected as suggested

Line 976: “Another element to consider is that the…” Figure 12: In the legend: “The homology model of TmGTase…”

Response: The text was corrected. We do not find “the homology model of TmGTase” in the Figure 12 legend, nor elsewhere. This protein has an experimentally determined 3D structure. This figure has been sent to supplementary materials.   

Reviewer 2 Report

Protein engineering on amylases and glucanotransferases has significant value in the industry. In order to identify the essential elements for hydrolysis and transglycosylation reaction , Rodrigo et al used computational approach with the enzymes in the GH13 family to generate contact maps, which is a kind of 2D representation of 3D structures, using a special algorithm. They compared each pair of residues in contact based on the conservation of coevolving residues. To increase of the transglycosylation activity, they selected several residues in TmAmyA (an amylase from Thermotoga maritima) and TmGTase (a glucanotransferase from Thermotoga maritima), produced the mutant enzymes and measured the activity in ratio of transglycosylation/hydrolysis activity.

I think this paper has a significant novelty with in-depth analysis of the contact residues essential for the activity. These days, the computational biology with powerful software and supercomputer is advancing fast, yielding accurate prediction of the protein structures. The computational calculation with aid of the artificial intelligence (AI) technology would surely make an impact on protein engineering in the near future. In this respect, research on structure and function relationship of the amylases based on the conservation of coevolving residues, using computational analysis described in this paper looks timely and appropriate. 

However, despite vast amount of data analysis, there is an important issue which was not addressed in this paper, the oligomerization status. All the analysis and protein designation are described without mentioning the oligomerization states of the enzymes. The active site and substrate binding groove of many enzymes in GH13 family are formed at the interface, particularly at the dimer interface between the N-domain and the catalytic domain. However, the distance measurement and analysis of the residue contact maps are described only based on the monmeric architecture. Therefore, this point should be properly addressed in the manuscript, or at least some interpretation with respect to the evolutionary approach should be included in the discussion part
Minor comments:
1.    In the experimental part the authors analyzed the enzyme   two different substrates: maltoheptaose for transglycosylation and starch for hydrolysis. If I now understand the analysis correctly, there are some concerns how to evaluate T/H ratio, even though the enrichment factor, Δf was presented. When different substrate used, the kinetic data should be expressed as “kcat/Km” which is not described in this paper.
2.    For the transglycosylation activity measurement a quantification with iodine reagent with maltoheptaose as substrate seems not to be the best method. Also the references are not adequately cited.

Author Response

Response to Reviewer 2

I think this paper has a significant novelty with in-depth analysis of the contact residues essential for the activity. These days, the computational biology with powerful software and supercomputer is advancing fast, yielding accurate prediction of the protein structures. The computational calculation with aid of the artificial intelligence (AI) technology would surely make an impact on protein engineering in the near future. In this respect, research on structure and function relationship of the amylases based on the conservation of coevolving residues, using computational analysis described in this paper looks timely and appropriate. 

Response: Thanks for the positive comments

  • However, despite vast amount of data analysis, there is an important issue which was not addressed in this paper, the oligomerization status. All the analysis and protein designation are described without mentioning the oligomerization states of the enzymes. The active site and substrate binding groove of many enzymes in GH13 family are formed at the interface, particularly at the dimer interface between the N-domain and the catalytic domain. However, the distance measurement and analysis of the residue contact maps are described only based on the monomeric architecture. Therefore, this point should be properly addressed in the manuscript, or at least some interpretation with respect to the evolutionary approach should be included in the discussion part

Response: We are conscious of the relevance of dimerization in this family of proteins. Oligomerization, besides being a stabilizing factor, can influence substrate and reaction specificity, and of course, it could be an extra element to be studied. In order to reduce the complexity of the system under investigation, all proteins, except TmGTase, Tm     AmyAa and 2VR5, were monomeric to the best of our knowledge. Even, TmGTase [https://doi.org/10.1107/S0907444901007740; Liebl, 10.1016/S0022-2836(02)00570-3]  and TmAmyA (doi:  10.1128/jb.179.3.941-948.1997) are      mostly a monomer in solution           . Thus, the possibility that oligomeric interphase could impact the determinants of specificity detected with the methodology proposed were due to the oligomeric interphase contribution is ruled out

Minor comments:

  1. In the experimental part the authors analyzed the enzym two different substrates: maltoheptaose for transglycosylation and starch for hydrolysis. If I now understand the analysis correctly, there are some concerns how to evaluate T/H ratio, even though the enrichment factor, Δf was presented. When different substrate used, the kinetic data should be expressed as “kcat/Km” which is not described in this paper.

Response: In fact, we did not make fine measurements of catalytic parameters. We were somewhat interested in comparing the hydrolysis and transglycosylation reactions relative to the wild-type protein. For the hydrolysis reactions, we only measured Vmax at saturating starch conditions (10 mg/mL), and no full Michaelis-Menten curves were run. Transglycosylation reactions, on the other hand, are more challenging to measure since once transglycosylation products start accumulating, they can be hydrolyzed by the enzyme. Thus, by measuring transfer to an alcohol, the events of transfer can be accounted for, even if the formed alkyl glycosides are later hydrolyzed at any intermediate glycosidic bond.

Mesuring transfer reaction for GTase did not have the same problem because it has a very poor (almost null) hydrolytic activity. The detection of growing amylose with Lugol using a short substrate, such as maltoheptaose, allowed a fast measurement of transglycsylation activity. For comparison purposes between variants, this measurement was enough.

  1. For the transglycosylation activity measurement a quantification with iodine reagent with maltoheptaose as substrate seems not to be the best method. Also the references are not adequately cited.

Response: Indeed the more accurate method to measure the transglycosidation activity would’ve been HPLC using isotope label substrates. But our methodology allows for a fast, easy and cheap comparison between the mutants and the wild type as a linear relationship is observed for the spectroscopic measurement of starch and starch equivalents when measured with iodine (Xiao, Z.; Storms, R.; Tsang, A. A quantitative starch-iodine method for measuring alpha-amylase and glucoamylase activities. Anal. Biochem. 2006, 351, 146–8; Zeeman, S.C.; Tiessen, A.; Pilling, E.; Kato, K.L.; Donald, A.M.; Smith, A.M. Starch Synthesis in Arabidopsis. Granule Synthesis, Composition, and Structure. Plant Physiol. 2002, 129, 516–529; Mould, D.L. Potentiometric and spectrophotometric studies of complexes of hydrolysis products of amylose with iodine and potassium iodide. Biochem. J. 1954, 58, 593–600)

Reviewer 3 Report

The manuscript describes a method for manipulating the hydrolase/transferase activity of GH13 enzymes. The method is based on the assumption that residues away from the catalytic site plays a role for defining the specificity. Here only structures are used for identifying the structural specificity determinants. The experimental work is not really supporting the method: It sounds like the mutations just affect the enzyme activity in general. At the end, the authors point at some of the limitations, which I was also thinking of while reading the manuscript: There might not be enough structures to include – and in particular transferase structures. Furthermore, the transferase/transglycosylation ability of many of the included enzymes is not known/well-characterised.

I found the manuscript interesting, but it is too long in my opinion. It becomes very exhausting to read, and some parts of the manuscript could be shortened. This would maybe also help the reader to see what is important.

I have a number of specific questions to the work:

You mention that Alpha-fold could become a game-changer when looking at structure/function relationship – and I agree. But you are using homology models created by Swiss Model – why not using AlphaFold?

Lines 74-84: I think that your description of two subgroups are not clear, since all GH13 enzymes could potentially do both reactions (hydrolysis or transglycolysis). In addition a minor thing: in the first line you define the abbreviation of glycoside hydrolase family 13 to GH-13 – but in another place (line 133) you defines it as GH13 (I think that the last one is the “correct”).

How did you select the 14 structures for Dataset 1? Why did you not include more transferases? Furthermore, as far as I can see, you have selected structures from many different subfamilies – is this on purpose? I would expect to see differences between subfamilies.

I would suggest that you in all tables in Suppl. material included information about subfamilies and organism of origin.

How do your findings correlate with the conserved regions of GH13?

Figure 1a: You have 3 green spots (transferases), but you have 4 transferases in our training set, where do the last one show up? I would expect that the 2VR5 would end-up in between the two groups of enzymes, as this enzyme has mixed activity (alpha-1,4-transferase and alpha-1,6-glucosidase).

When you do your analysis do you include the full structure, or only the catalytic domain?

You write AmyA in a few places (e.g. line 237 and 254) do you mean TmAmyA or is it another AmyA?

Lines 253-255: So you conclude that TmAmyA is closet to TmGTase among all the structures you have included in your analysis? Is this surprising or what you expected? Are we then down to organism origin playing a larger role than actual specificity?   

I would have liked to see a table, where the hydrolase and transferase activities were given for all variants included in the manuscript. It would also help with the readability, as it would also give an overview of which variants you made.

How was the thermal stability of the two enzymes affected by the introduction of the mutations? You mention that one of the areas mutated contains a calcium binding site. In GH13 calcium can be important for the structural integrity. Could it be a problem that you use 70C heating as a purification step, if the mutations are affecting structural integrity?

Line 368: You mention that residue H222 has been shown previously to be important for increasing transglycosylation/hydrolysis ratio – did you also see this residue showing up in your analysis?

Line 449: What do you mean by “spurious”?

Does it make sense to do MD simulations on a homology model?

Figure 8: There is a residue shown in light pink – what is this? The transition state stabilizer?

Lines 721-724: I am missing a reference for this section.

Lines 845-853: I do not understand why you are comparing your results with results from significantly other types of enzymes.

What template did you use for modelling TmAmyA?

Where did you get the TmAMyA plasmid/DNA from? You only describes how TmGTase DNA was obtained?

How were your protein concentration determined?

How do you define “U” in relation to the TmAmyA? You most have done a specific activity measurement in order to determine the U – how was that done?

Minor comment: You write “bacillar” in a few places do you mean “bacterial” (i.e. from bacteria) or bacilli (i.e. from Bacillus)?

Author Response

Response to Reviewer 3

  1. I found the manuscript interesting, but it is too long in my opinion. It becomes very exhausting to read, and some parts of the manuscript could be shortened. This would maybe also help the reader to see what is important.

Response: The presentation of results and analysis of Molecular Dinamics has been reduced considerably. Also many fugures were sent to Supplementary Material to make the manuscript easier to read.

I have a number of specific questions to the work:

  1. You mention that Alpha-fold could become a game-changer when looking at structure/function relationship – and I agree. But you are using homology models created by Swiss Model – why not using AlphaFold?

Response. Alpha-fold was reported after the design and construction of the variants reported in this work. However, we have compared a couple of structures predicted with Alpha-fold and with Swiss-Model and are superimposable. For example, the RMSD across all pairs of atoms of 0.829 Å was calculated for modelled TmAmyA with Swiss-Model and AlphaFold2.

  1. Lines 74-84: I think that your description of two subgroups are not clear, since all GH13 enzymes could potentially do both reactions (hydrolysis or transglycolysis). In addition a minor thing: in the first line you define the abbreviation of glycoside hydrolase family 13 to GH-13 – but in another place (line 133) you defines it as GH13 (I think that the last one is the “correct”).

Response: You are right, enzymes in GH13 family show in higher or lower degree both reactions. However there are some whose reaction specificity preference is clearly defined. Those are the ones we used to define the limits. Anyways we removed this arbitrary classification and tried to clarify this point in the text. Also we changed all the abbreviations GH-13 to GH13 in the text. Thanks for noticing,

  1. How did you select the 14 structures for Dataset 1? Why did you not include more transferases? Furthermore, as far as I can see, you have selected structures from many different subfamilies – is this on purpose? I would expect to see differences between subfamilies.

Response: We selected only the structures of enzymes bound to acarbose, a transition state analog, to represent the proteins in their active conformation. In that sense, transferases were less represented in the PDB. we clarified this in the text in the second line of the Results section,

  1. I would suggest that you in all tables in Suppl. material included information about subfamilies and organism of origin.

Response:. Subfamily of CAZy and organism of origin were added for each protein in Supplementary material.

  1. How do your findings correlate with the conserved regions of GH13?

Response: Within conserved regions, some positions show variability among enzymes, especially related to their substrate or reaction specificity. Our findings detect these residues that are part of the substrate-binding site and target mutagenesis in several works reported in the literature. The main idea of the proposed  methodology is to cover residues not necessarily near the active site. Thus, in this work, we only explored residues 274and 279 in one of the highly conserved regions of the GH13 family. Interestingly, albeit being part of the conserved region IV, this position shows a pattern that depends on the type of enzyme. Thus, amylases prefer Val at this position, while CGTases, Ile and, maltogenic amylases, Leu. Residue 279 is right after the same conserved region and also shows differential amino acid preferences: Met in CGTases, Thr for maltogenic amylases, and Asn for amylases, which was the amino acid proposed by the algorithm.

The other positions explored in GTase are not part or near the highly conserved regions in this family of proteins. Phe 71 and Val 86 are in helix 2 and at the bottom of β-strand 3, respectively. Both in the interphase between β strands and É‘-helices in the TIM barrel.

The unintended mutations E77G and E226K are in the outer part of helices 2 and 5 of the TIM barrel, respectively, and are mostly exposed to the solvent.

Regarding the mutations carried out on TmAmyA, they locate at the very long loop joining β-strand 2 and helix-2 in the TIM barrel

  1. Figure 1a: You have 3 green spots (transferases), but you have 4 transferases in our training set, where do the last one show up? I would expect that the 2VR5 would end-up in between the two groups of enzymes, as this enzyme has mixed activity (alpha-1,4-transferase and alpha-1,6-glucosidase).

Response: There are 4 transferases but two of them are near each other (upper points).

  1. When you do your analysis do you include the full structure, or only the catalytic domain?

Response. We include the full structure, but after comparing the contacts with their reference (with only domains A, B and C). The method could not analyze contacts not belonging to these domains and, therefore, would not appear in the enrichment analysis.

  1. You write AmyA in a few places (e.g. line 237 and 254) do you mean TmAmyA or is it another AmyA?

Response: It is the same. Thanks for noticing. All AmyA have been changed to TmAmyA.

  1. Lines 253-255: So you conclude that TmAmyA is closet to TmGTase among all the structures you have included in your analysis? Is this surprising or what you expected? Are we then down to organism origin playing a larger role than actual specificity?   

Response: For these two enzymes, it seems to be the case.. It depends whether the enzymes diverge before or after speciation. Enzymes in the same organism drift together under the same environmental pressure.

  1. I would have liked to see a table, where the hydrolase and transferase activities were given for all variants included in the manuscript. It would also help with the readability, as it would also give an overview of which variants you made.

Response:Two tables were included in the manuscript, one with all TmAmyA variants and another with TmGTase variants.

  1. How was the thermal stability of the two enzymes affected by the introduction of the mutations? You mention that one of the areas mutated contains a calcium binding site. In GH13 calcium can be important for the structural integrity. Could it be a problem that you use 70 °C heating as a purification step, if the mutations are affecting structural integrity?

Response: TmAmyA and TmGTase are thermostable enzymes obtained from a hyperthermophile bacterium Thermotoga maritima. Although the stability of mutants was not experimentally determined, the detection of biological activity is evidence of the stability of these proteins. Nevertheless, the objective of the experiment was to determine the ratio of transglycosylation and hydrolysis employing the same quantity of protein for both activity assays.

  1. Line 368: You mention that residue H222 has been shown previously to be important for increasing transglycosylation/hydrolysis ratio – did you also see this residue showing up in your analysis?

Response: H222Q shows up but within the limits of what we considered to selected a variant.Iit was not elegible (-0.05); however, it was interesting to observe the analysis suggested leucine (0.25) as the amino acid favoring transnglycosidation, a substitution not explored yet.

  1. Line 449: What do you mean by “spurious”?

Response:These are unintended mutations that eventually occur due to the low fidelity of the Taq-polymerase.

  1. Does it make sense to do MD simulations on a homology model?

Response: TmAmyA and its triple mutant (K98P/D99A/H222Q) share high identity with the Thermotoga petrophila amylase, used as a template for the model, with only 6 substitutions. Also, in other works (Sefidbakht et al.,. J. Biomol. Struct. Dyn. 2016, 35, 574–584; Hasmaliana et al.,. J. Mol. Graph. Model. 2016, 67, 1–13; 1. Biswas et al., Soft Matter 2020, 16, 3050–3062) MD with homology models have been accomplished taking in account the limitations of the obtained results.

  1. Figure 8: There is a residue shown in light pink – what is this? The transition state stabilizer?

Response: Yes, it is the Aspartic acid that stabilizes the transition state. It is clarified now in the figure legend.

  1. Lines 721-724: I am missing a reference for this section.

Response: A reference has been added.

  1. Lines 845-853: I do not understand why you are comparing your results with results from significantly other types of enzymes.

Response: Comparison with more closely related enzymes has been provided.

The mentioned enzymes in this part of the Discussion were the β-glucosidase from Thermotoga neapolitana and almond Beta-Glucosidase, which belong to the family GH1 (CAZy), and also the α-L-arabinofuranosidase from Thermobacillus xylanilyticus (GH51).  The two enzymes used as models (TmAmyA and TmGTase from family GH13) have differences with enzymes from GH1 and GH51 families such as substrate specificity and stereochemistry of O-glycosidic bond. Nevertheless, the comparison was accomplished because the effect of flexibility and dynamical events in protein specificity or activity have been analyzed for these enzymes of GH1 and GH51 families. In spite of the differences of GH13 family with the other two families, both types of enzymes are associated with the reaction of hydrolysis and/or transglycosydation. Another reason was that all these enzymes share certain structural and functional properties with the GH13 family such as a (β/α) 8 barrel and the double-displacement mechanism including a retaining of the configuration in the final products. The comparison with the Amylosucrase from Deinococcus geothermalis (DGAS) -a member of GH13 family- also was added in the manuscript.

  1. What template did you use for modelling TmAmyA?

Response: The template used for modelling TmAmyA was the crystallographic structure of amylase from Thermotoga petrophila (PDB ID 5M99). These proteins have a 98 % of sequence identity.

  1. Where did you get the TmAMyA plasmid/DNA from? You only describes how TmGTase DNA was obtained?

Response: The cloning of AMyA gene was previously reported in Damian-Almazo, et al., Appl. Environ. Microbiol. 2008, 74, 5168–5177, doi:10.1128/AEM.00121-08. It is clarified now in the Methodology section.

  1. How were your protein concentration determined?

Response: In all cases protein concentration was accomplished by the Bradford method using a kit from ThermoFisher Scientific following the instructions of the manufacturer. This information was added to the Methods section.

  1. How do you define “U” in relation to the TmAmyA? You most have done a specific activity measurement in order to determine the U – how was that done?

Response: A unit of enzyme activity is defined as the amount of enzyme required to release one ?mol of glucose equivalents per min and is reported in terms of mg of AmyA.

  1. Minor comment: You write “bacillar” in a few places do you mean “bacterial” (i.e. from bacteria) or bacilli (i.e. from Bacillus)?

Response: We refer to Bacilli, from Bacillus. We made the corresponding corrections. Some of the microorganisms (genus Niallia and Geobacillus, for example), were once classified as Bacillus due to their similarity.

Reviewer 4 Report

The manuscript aims at finding ways to modulate the ratio of transglycosylation and hydrolysis reactions catalysed by family GH13 glycoside hydrolases. It has a strong focus on describing a methodology to select suitable mutations. The wet experiments are relatively few and the results are not so impressive. This raises some doubts concerning the usefulness of the methodology. Anyway, modulation of transfer/hydrolysis ratios is of great interest and has been proven before to be challenging, so publication of the study in some form can be motivated.

My main suggestion is to shorten the manuscript to about half its current length. The scientific value of the study does not justify a longer manuscript. Additional parts can maybe be moved to supplementary information.

Additional comments:

The abstract must contain quantitative data from the wet experiments.

Why were the hydrolysis and transfer rates measured in different ways in the two sets of mutants? Especially the transfer rates were measured in two totally different reactions. In fact, it would have been highly interesting to test each enzyme variant in all the reactions. That would have given interesting information on potential differences in the transfer activity due to the use of alcohol or carbohydrate acceptors.

Author Response

Response to Reviewer 4

  1. My main suggestion is to shorten the manuscript to about half its current length. The scientific value of the study does not justify a longer manuscript. Additional parts can maybe be moved to supplementary information.

Response: The manuscript was significantly shortened, and the changes can be viewed as control changes in MS Word. 

Additional comments:

  1. The abstract must contain quantitative data from the wet experiments.

Response: Quantitative data has been included in the abstract.

  1. Why were the hydrolysis and transfer rates measured in different ways in the two sets of mutants? Especially the transfer rates were measured in two totally different reactions. In fact, it would have been highly interesting to test each enzyme variant in all the reactions. That would have given interesting information on potential differences in the transfer activity due to the use of alcohol or carbohydrate acceptors.

Measuring transglycosylation reaction in TmAmyAis challenging since once transglycosylation products start accumulating, they can be hydrolyzed by the enzyme. Thus, by measuring transfer to an alcohol, the events of transfer can be accounted for, even if the formed alkyl glycosides are later hydrolyzed at any intermediate glycosidic bond.

Mesuring transfer reaction for GTase did not have the same problem because it has a very poor (almost null) hydrolytic activity. The detection of growing amylose with Lugol using a short substrate, such as maltoheptaose, allowed a fast measurement of transglycsylation activity. For comparison purposes between variants, this measurement was enough.

We have tested the alcoholysis reaction with TmGTase, but no alkyl glycosides were observed.

Round 2

Reviewer 1 Report

Most of the previous questions have been adequately handled by the authors.

What remains to be improved in my opinion, is to describe the 4 datasets when they appear for the first time in the Results. For example, datasets 2 and 3 are mentioned in lines 167 and 203, but there it is not explained what they contain. The reader has to go to the supplemental information to find out.

Furthermore, the grammar needs to be improved throughout the manuscript in order to improve readability (e.g., punctuation, half sentences).

Some remaining specific points:

  1. The (new) Table 2 could benefit from adding T/H or H/T ratios since these are discussed in the accompanying text.
  2. "Arquea" should be "Archaea" 

Author Response

Response to Reviewer 1

  1. What remains to be improved in my opinion, is to describe the 4 datasets when they appear for the first time in the Results. For example, datasets 2 and 3 are mentioned in lines 167 and 203, but there it is not explained what they contain. The reader has to go to the supplemental information to find out.

Response: A description of every Dataset has been added in the corresponding Results Section.

  1. Furthermore, the grammar needs to be improved throughout the manuscript in order to improve readability (e.g., punctuation, half sentences).

Response: An English native speaker has gone through the manuscript and made significant improvements. We hope the manuscript is written correctly and is more readable now.

Some remaining specific points:

  1. The (new) Table 2 could benefit from adding T/H or H/T ratios since these are discussed in the accompanying text.

Response: The hydrolysis/transglycosidation (H/T) ratio has been added in table 2

  1. "Arquea" should be "Archaea"

Response: Thanks for noticing. This mistake has been corrected

Reviewer 2 Report

The revised manuscript can be accepted as a regular article for this journal.

Author Response

Response to Reviewer 2

Comment:

  1. The article can be accepted as in present form. In future work, the kinetic analysis may help to much more strengthen your modelling study.

Response: Thank you for your valuable comments. We certainly will consider all the suggestions about a more formal way to measure kinetic activity for transfer reactions in the future characterization of new variants.

Reviewer 3 Report

I am happy to see that you have replied to all my questions.

I think the manuscript has been significantly improved - and I found it much easier to read the text. I especially like seeing the actual activity results in tables 1 and 2, and that you address the conserved regions in GH13.

I only found minor (typing) mistakes. Here are a few examples (not an exhaustive list): line 433: ", as." or line 514, where a link to a reference has gone wrong. You also use the words "Arqueas'" and "arqueal", I believe that this are Mexican/Spanish words referring to "Archaea"?

Author Response

Response to Reviewer 3

.

  1. I think the manuscript has been significantly improved - and I found it much easier to read the text. I especially like seeing the actual activity results in tables 1 and 2, and that you address the conserved regions in GH13.

Response: Thank you for your valuable comments

  1. I only found minor (typing) mistakes. Here are a few examples (not an exhaustive list): line 433: ", as." or line 514, where a link to a reference has gone wrong.

Response: These typing mistakes, as well as some grammar mistakes, have been corrected. An English native speaker has gone through the manuscript and made significant improvements. We hope the manuscript is written correctly and is more readable now.

  1. You also use the words "Arqueas'" and "arqueal", I believe that this are Mexican/Spanish words referring to "Archaea"?

Response: Thanks for noticing. This mistake has been corrected.

Reviewer 4 Report

I find the revised version acceptable for publication

Author Response

Thank you for your valuable comments.

Round 3

Reviewer 2 Report

The article can be accepted as in present form. In future work the kinetic analysis may help to much more strengthen your modelling study.

Author Response

(The authors gave the same response as above.)
